# The Antimicrobial Peptide Cathelicidin Exerts Immunomodulatory Effects via Scavenger Receptors

**DOI:** 10.3390/ijms24010875

**Published:** 2023-01-03

**Authors:** Ryo Amagai, Toshiya Takahashi, Hitoshi Terui, Taku Fujimura, Kenshi Yamasaki, Setsuya Aiba, Yoshihide Asano

**Affiliations:** Department of Dermatology, Tohoku University Graduate School of Medicine, Sendai 980-8574, Japan

**Keywords:** cathelicidin, scavenger receptors, damage-associated molecular patterns (DAMPs), psoriasis, innate immunity

## Abstract

An active form of cathelicidin antimicrobial peptide, LL-37, has immunomodulatory and stimulatory effects, though the specific pathways are not clear. The purpose of this study was to identify the cellular pathways by which LL-37 amplifies the inflammation induced by damage-associated molecular patterns (DAMPs). We performed DNA microarray, reverse transcription polymerase chain reaction, immunoblotting, and proximity ligation assays using cultured keratinocytes treated with LL-37 and/or the DAMP poly(I:C), a synthetic double-stranded RNA. In contrast to the combination of LL-37 and poly(I:C), LL-37 alone induced genes related to biological metabolic processes such as VEGFA and PTGS2 (COX-2). Inhibition of FPR2, a known receptor for cathelicidin, partially suppressed the induction of VEGFA and PTGS2. Importantly, VEGFA and PTGS2 induced by LL-37 alone were diminished by the knockdown of scavenger receptors including SCARB1 (SR-B1), OLR1 (SR-E1), and AGER (SR-J1). Moreover, LL-37 alone, as well as the combination of LL-37 and poly(I:C), showed proximity to the scavenger receptors, indicating that LL-37 acts via scavenger receptors and intermediates between them and poly(I:C). These results showed that the broad function of cathelicidin is generally dependent on scavenger receptors. Therefore, inhibitors of scavenger receptors or non-functional mock cathelicidin peptides may serve as new anti-inflammatory and immunosuppressive agents.

## 1. Introduction

Antimicrobial peptides (AMPs) play an essential role in the immune defense of almost all plants and animals [1]. The AMP cathelicidin was the first identified in mammalian skin [2]. Human cathelicidin protein is transcribed as an inactive form and is proteolyzed by serine proteases into multiple active cathelicidin peptides, including LL-37 [3]. Various types of cells, such as epidermal keratinocytes, intestine cells, respiratory epithelial cells, neutrophils, T cells, natural killer cells, monocytes, and mast cells express LL-37 [4,5,6]. LL-37 has intrinsic antimicrobial activity [7,8] as well as immunomodulatory and stimulatory effects [9]. It acts on host cells through cytokine and chemokine production [10], leading to the chemotaxis of neutrophils, monocytes, and lymphocytes [11,12], in addition to angiogenesis [13,14]. Therefore, LL-37 is implicated in the pathogenesis of various human diseases, including inflammatory bowel disease [15], lung cancer [16], asthma, cystic fibrosis, chronic obstructive pulmonary disease [17], Alzheimer’s disease [18], systemic sclerosis [19], systemic lupus erythematosus, rheumatoid arthritis, atherosclerosis [20], rosacea [21,22], atopic dermatitis [9], and psoriasis [23]. In many of these disorders, the presence of excess LL-37 is thought to enhance the local tissue inflammatory response.

In keratinocytes, LL-37 induces proinflammatory cytokines such as interleukin (IL)-8 and IL-18 [24,25], and cyclooxygenase (COX)-2 [26]. The function of LL-37 has been linked to its membrane activity. LL-37 activates G-coupled receptors such as formyl peptide receptor 2 (FPR2, FPRL1) [12] and enables extracellular nucleic acids to enter the cytosol [27]. Furthermore, Di Nardo et al. showed that LL-37 differentially alters membrane receptor mobility and structures in dendritic cells and keratinocytes [28]. LL-37 also binds to double-stranded RNA (dsRNA), a damage-associated molecular pattern (DAMP) that represents viral molecules, and is efficiently taken up into cells together with dsRNA by Toll-like receptors (TLRs), which results in intracellular RNA sensors inducing inflammatory cytokines such as interferon (IFN)-β. We have demonstrated that LL-37 enables keratinocytes and macrophages to recognize self-non-coding RNA, a DAMP, by facilitating its binding to cell surface scavenger receptors and intracellular pathogen recognition receptors (PRRs) in vitro and in human psoriatic skin [29,30]. However, it is not clear if LL-37 alone and the combination of LL-37 and dsRNA facilitate their function using the same receptor and signaling pathways or use other specific pathways by which they exert their immunomodulatory effects. In this study, we comprehensively investigated the mechanism by which LL-37 alone and the combination of LL-37 and dsRNA enter the cytoplasm via the cytomembrane and invoke inflammation and other immunological functions.

We performed a DNA microarray, RT-PCR, immunoblotting, and a proximity ligation assay (PLA) using cultured keratinocytes with LL-37 and/or poly(I:C), a synthetic double-stranded RNA. LL-37 alone induced genes mainly related to metabolism and biological processes, including COX-2 and vascular endothelial growth factor (VEGF)-A. In contrast, LL-37 complexed with poly(I:C) induced immune and inflammation-related genes such as retinoic acid-inducible gene-I (RIG-I), tumor necrosis factor (TNF)-α, C-X-C motif chemokine ligand 10 (CXCL10), and IFN-β1. Interestingly, the inhibition of scavenger receptors abolished the uptake of LL-37 alone and subsequent gene induction, comparable to the combination of LL-37 and poly(I:C). Contrary to our expectation, the inhibition of FPR2 did not change the uptake of LL-37 alone nor expression of the genes. These results highlight that both LL-37 alone and the combination of LL-37 and dsRNA pass primarily through scavenger receptors, and that LL-37 induces inflammation through various pathways depending on the presence or absence of coexisting DAMPs. These results imply that the inhibition of scavenger receptors is a possible new therapeutic intervention to block various types of LL-37-related inflammation with or without DAMPs.

## 2. Results

### 2.1. Unlike the Combination of LL-37 and dsRNA, LL-37 alone Induces a Set of Genes Mainly Related to Biological Metabolic Processes in Keratinocytes

First, to see how keratinocytes responded to the combination of nucleic acids and LL-37, we added poly(I:C) or two types of double-stranded DNA (poly(dA:dT) and CpG ODN) alone or combined with LL-37 to normal human epidermal keratinocytes (NHEKs). The combination of LL-37 and poly(I:C) induced *IL6* mRNA, whereas the combination of LL-37 and poly(dA:dT) induced less than poly(I:C), and the combination with CpG ODN did not induce *IL6* mRNA (Figure 1a,b). Since dsRNA reacted more strongly in NHEKs than dsDNA in combination with LL-37, we then stimulated NHEKs with or without poly(I:C) with several concentrations of LL-37 to test their cytokine induction. Although poly(I:C) alone had almost no effect at the concentrations tested, LL-37 in combination with poly(I:C) significantly increased the expressions of *IL6*, *IL36G*, and *IFNB1* mRNAs (Figure 1c–e). We next examined the mRNA expression induced by LL-37 and poly(I:C) alone or in combination using a DNA microarray to identify the characteristics of the gene sets induced by each stimulus (Figure 1f) and identified 1192 genes that were induced two-fold or more by adding LL-37 alone. Only 53 genes were upregulated more than two-fold by single poly(I:C) treatment. Furthermore, 458 genes showed a four-fold or greater increase in expression upon treatment with LL-37 and poly(I:C) compared to LL-37 or poly(I:C) alone. Moreover, of the 1192 genes, 94 genes were upregulated more than two-fold upon treatment with LL-37 alone and were further upregulated following treatment with LL-37 and poly(I:C). Gene ontology (GO) analysis revealed that LL-37 alone significantly upregulated biological and metabolic processes (Figure 1g), whereas combined stimulation with LL-37 and poly(I:C) significantly induced immune defense responses and type I IFN processes (Figure 1h). These results suggested that compared to LL-37 alone, LL-37 in combination with dsRNA produced completely distinct signals and should have different immunological roles in vitro. To further this analysis, we identified genes that were induced by LL-37 alone but remained almost unchanged when poly(I:C) was added. We also identified genes that were upregulated by the combination of LL-37 and poly(I:C) but not by LL-37 or poly(I:C) alone. A heatmap (Figure 1i) indicates the top genes with the highest variation. *VEGFA* and *prostaglandin-endoperoxide synthase 2 (PTGS2)* were selected as representative genes among those related to biological and metabolic processes in the group of genes induced by LL-37. *DExD/H-box helicase 58* (*DDX58*), *TNFA*, *CXCL10*, and *IFNB1* were chosen from genes associated with immune processes whose expression was increased by the combination of LL-37 and poly(I:C). VEGF-A is an important regulator of angiogenesis and a pathogenetic factor in inflammatory diseases and solid tumors [31]. *PTGS2*, also called COX-2, is an enzyme that converts arachidonic acid to prostaglandin H2 (PGH2). COX-2 is rapidly inducible in response to numerous intracellular and extracellular stimuli, and acts in a pro-inflammatory fashion [32]. TNF-α was originally described as a circulating factor that can cause the necrosis of tumors, but has been also identified as a key regulator of the inflammatory response [33]. CXCL-10, also called IFN γ-inducible protein 10 (IP-10), binds to CXC chemokine receptor (CXCR)-3 and regulates immune responses by the activation and recruitment of leukocytes, such as T cells, eosinophils, monocytes, and NK cells [34]. IFN-β is a type I IFN that binds to a heterodimeric transmembrane receptor composed of the subunits IFNAR1 and IFNAR2, and induces IFN-stimulated genes [35]. *DDX58* is the gene for RIG-I. RIG-I, as well as melanoma differentiation-associated gene 5 (MDA5), is a RIG-I-like receptor (RLR) that plays a major role in the pathogen sensing of RNA virus infections to initiate and modulate antiviral immunity [36]. RT-PCR analysis confirmed that these genes were induced (Appendix A). These results suggest that LL-37 alone induces a set of genes mainly related to biological metabolic processes in keratinocytes, unlike the combination of LL-37 and dsRNA, which induces immune defense responses and type I IFN processes.

### 2.2. FPR2 Is a Receptor for LL-37 alone in Keratinocytes, but Is Not Involved in Signal Transmission of the Combination of LL-37 and poly(I:C)

Several membrane receptors for LL-37 have been reported in keratinocytes, such as FPR2, P2X7 receptor (P2X7R), and epidermal growth factor receptor (EGFR) [12,25,37,38]. We therefore investigated the receptor for LL-37 by adding LL-37, poly(I:C), and various inhibitors against known receptors to keratinocytes. We treated keratinocytes with WRW4, a known FPR2 inhibitor, followed by stimulation with LL-37, and found that the induction of *PTGS2* and *VEGFA* mRNA by LL-37 was partially inhibited by WRW4 (Figure 2a,b). In contrast, *CXCL10* and *IFNB1* mRNA induced by the combination of LL-37 and poly(I:C) were not suppressed by WRW4 (Figure 2c,d). The induction of VEGF-A by LL-37 was confirmed by enzyme-linked immune-sorbent assay (ELISA) using culture supernatants. Similar to *VEGFA* mRNA, the induction of VEGFA protein by LL-37 alone was partially inhibited by WRW4, but not by the combination of LL-37 and poly(I:C) (Figure 2e). As with mRNA, CXCL10 protein was also induced by the combination of LL-37 and poly(I:C) and was not inhibited by WRW4 (Figure 2f). The gene induced by LL-37 alone was dose-dependently inhibited by WRW4 (Appendix A). Similar experiments were performed with Boc-MLF, an FPR1 inhibitor, and KN-62, a P2X7R antagonist, but these did not inhibit or only slightly inhibited the induction of *PTGS2* and *VEGFA* mRNA by LL-37 alone, or *CXCL10* and *IFNB1* mRNA by the combination of LL-37 and poly(I:C) (Figure 2g–j).

LL-37 has been reported to induce inflammatory mediators by phosphorylating p38 via FPR2 in neutrophils [39]. Therefore, we evaluated the phosphorylation of p38 using Western blotting and found that p38 was also phosphorylated in keratinocytes upon LL-37 stimulation, and that the phosphorylation of p38 was not changed by the addition of poly(I:C). Moreover, p38 phosphorylation by LL-37 alone was partially inhibited by treatment with WRW4 and an endocytosis inhibitor, Pitstop 2 (Figure 2k). These results suggest that FPR2 acts as a receptor for LL-37 alone, inducing *VEGFA* and *PTGS2* via p38 phosphorylation, but not for poly(I:C) or the combination of LL-37 and poly(I:C).

### 2.3. LL-37 alone and in Combination with poly(I:C) Is Transported by Scavenger Receptors in Keratinocytes

We previously reported that complexes of LL-37 and dsRNA are taken up via scavenger receptors and sensed by receptors in the cytoplasm and endosomes [29]. Actually, the cytokine induction by LL-37 alone was attenuated by fucoidan, an inhibitor of multiple scavenger receptors (Appendix A). Next, to investigate whether scavenger receptors are also required for LL-37 alone to act, and which scavenger receptors are important, we knocked down various scavenger receptors using small interfering RNAs (siRNAs) in NHEKs and examined their effects on mRNA induction. Of the various types of scavenger receptors, we selected those for knockdown that were highly expressed in NHEKs or induced by LL-37 and poly(I:C) in the DNA microarray (Appendix A): namely, SR-A3, B1, D1, E1, and J1. Oxidized low density lipoprotein receptor 1 (*OLR1*, SR-E1) and cluster of differentiation 68 (*CD68*, SR-D1) were upregulated in these scavenger receptors by combined stimulation with LL-37 and poly(I:C). The suppression of mRNA expression of the targeted scavenger receptors by siRNAs was confirmed to be sufficient (Appendix A). Genes induced by LL-37 alone (*VEGFA* and *PTGS2*) were inhibited by knockdown of various scavenger receptors, especially by knockdown of SR-E1, SR-B1 (*SCARB1*), and SR-J1 (*AGER,* RAGE) (Figure 3a,b). For genes induced by the combination of LL-37 and dsRNA (*CXCL10*, *IFNB1*, and *TNFA*), siRNA of SR-B1 reduced the induction of *IFNB1* and *TNFA* mRNAs. In addition, the inhibition of SR-E1 expression suppressed *TNFA* mRNA (Figure 3c–e). However, other siRNAs against other scavenger receptors did not show significant inhibition of those genes induced by the combination of LL-37 and dsRNA.

Next, we performed PLA to evaluate the binding of LL-37 to these scavenger receptors. In the PLA, two targeted proteins can be detected as a fluorescence spot when they are in close proximity (within 40 nm). PLA was performed using antibodies against the scavenger receptors targeted in the knockdown experiments and anti-LL-37 antibody. The proximity of LL-37 to the scavenger receptors was detected in combination with poly(I:C), as previously reported. In addition, we newly discovered that LL-37 alone also showed proximity to the scavenger receptors. In particular, a large number of PLA spots were detected using the combination of SR-E1 and LL-37 (Figure 3f). These results indicate that scavenger receptors also act as receptors for LL-37 alone on the surface of keratinocytes.

### 2.4. The Combination of LL-37 and poly(I:C) Was Recognized by TLRs on Endosomes and Cytosolic RNA Sensors, Followed by Clathrin-Dependent Endocytosis and Cytokine Induction

LL-37 and dsRNA complexes have been reported to be taken up into cells by endocytosis and sensed by TLR3 on endosomes and by the RIG-I/MDA5/MAVS system, a cytosolic RNA sensor [29,30]. Here, we investigated the effect of endocytosis inhibitors on chemokine induction by LL-37 alone or in combination with poly(I:C). Pitstop 2, an inhibitor of clathrin-dependent endocytosis, almost completely inhibited *CXCL10, IFNB1*, *TNFA*, and *DDX58* induced by LL-37 and poly(I:C), and significantly inhibited *PTGS2* mRNA induced by LL-37 alone. The induction of *VEGFA* mRNA by LL-37 alone had a tendency to be suppressed by Pitstop 2, but not significantly (Figure 4a–f). The induction of these genes by LL-37 alone and by the combination of LL-37 and poly(I:C) was also inhibited by Pitstop 2 when protein levels of TNF-α and CXCL-10 in supernatants were measured by ELISA (Figure 4g,h). These results suggest that LL-37 alone as well as the complex of LL-37 and poly(I:C) stimulates cells via clathrin-dependent endocytosis. When the transfection reagent Lipofectamine 3000 was combined with poly(I:C) and added to NHEK, it induced *IL6* mRNA. However, the induction was less than that by the combination with LL-37, suggesting that poly(I:C) acted by entering the cells (Appendix A). In addition, when we stimulated NHEKs with a high concentration of poly(I:C), poly(I:C) alone induced *CXCL10, TNFA*, and *DDX58*, but not *IFNB1* mRNA (Appendix A). While LL-37 lowered the concentration threshold at which poly(I:C) acted, *IFNB1* mRNA was induced only when LL-37 and poly(I:C) were combined.

Next, we examined the intracellular signaling pathways for LL-37 alone or the combination of LL-37-poly(I:C) for stimulating keratinocytes. Bafilomycin A1 (Baf-A1) is a vacuolar type H(+)-ATPase inhibitor that inhibits endosomal receptors, including TLR3 [40,41]. Treatment with Baf-A1 partially inhibited *CXCL10*, *IFNB1*, *TNFA*, and *DDX58* mRNAs induced by the combination of LL-37 and poly(I:C), but had little effect on *PTGS2* and *VEGFA* mRNA induced by LL-37 (Figure 4a–f). Since TLR3 is known to be a receptor for dsRNA in endosomes, this result suggests that TLR3 is involved in the response to the combination of LL-37 and poly(I:C), as previously reported [29]. We further used siRNA to knock down TLR3 and MAVS in NHEKs and evaluated their responses to LL-37 and poly(I:C). Immunoblotting confirmed that TLR3 and MAVS siRNAs sufficiently knocked down their target proteins after 24 h of transfection (Appendix A). Stimulation with a combination of LL-37 and poly(I:C) induced *TLR3* mRNA in the control group, but the respective mRNAs were sufficiently suppressed in the group transfected with siRNAs against TLR3 and/or MAVS (Appendix A). Knockdown of either TLR3 or MAVS inhibited the induction of *CXCL10*, *IFNB1*, and *TNFA* mRNA by the combination of LL-37 and poly(I:C) (Figure 4i–k). Furthermore, knockdown of both TLR3 and MAVS resulted in almost complete inhibition. 

DsRNA is reportedly recognized by TLR3 and the RIG-I/MDA5/MAVS system in keratinocytes in combination with LL-37, resulting in phosphorylation of TANK binding kinase 1 (TBK1) and interferon regulatory factor 3 (IRF3) and the induction of cytokines such as type I IFNs [30]. Immunoblot analysis indicated the phosphorylation of TBK1 and IRF3 upon combined stimulation with LL-37 and poly(I:C). Clathrin inhibition by Pitstop 2 decreased the phosphorylation of TBK1 and IRF3 by LL-37 and poly(I:C), indicating the involvement of clathrin-dependent endocytosis. In contrast, the phosphorylation of TBK1 and IRF3 was not attenuated by WRW4, an FPR2 inhibitor, suggesting again that FPR2 is not involved in LL-37 and poly(I:C) endocytosis (Figure 4l). In addition, the phosphorylation of p38 by LL-37 was poorly suppressed by Pitstop 2, suggesting that clathrin-dependent endocytosis is involved when LL-37 and poly(I:C) are combined (Figure 2k). These results suggest that cytokine induction by the combination of LL-37 and poly(I:C) is mediated by clathrin-dependent endocytosis, binding to TLR3 and RIG-I/MDA5/MAVS, and TBK1 and IRF3 phosphorylation.

## 3. Discussion

The antimicrobial peptide LL-37 has not only well-known antibacterial [7], antifungal [8], and antiviral [42] effects, but also induces various immune responses via the secretion of cytokines and chemokines [10], as well as stimulating chemotaxis [11,12] and angiogenesis [14] in host cells. In addition, LL-37, which is cationic and amphiphilic, can bind to dsDNA and dsRNA and facilitate their intake into the cytoplasm of host cells, followed by the induction of type I IFN and TNF-α [30,43]. In this study, we comprehensively investigated the effects of LL-37 on keratinocytes by comparing the effects of LL-37 alone with those of LL-37 in the presence of dsRNA.

We first showed that LL-37 alone induces a set of genes mainly related to biological metabolic processes in keratinocytes, and these genes are different from those induced by the combination of LL-37 and dsRNA. Using GO analysis, we demonstrated that LL-37 alone induces genes involved in biological and metabolic processes such as *VEGFA* and *PTGS2* (COX-2) in keratinocytes. VEGF-A has been implicated in the pathogenesis of psoriasis [44]. For example, *Vegfa*-transgenic mice exhibit psoriasis-like skin inflammation [45], which is improved by the inhibition of VEGF-A [46]. Furthermore, the use of bevacizumab, a VEGF monoclonal antibody, has been reported to improve skin manifestations in patients with psoriasis [47]. COX-2 expression has also been shown to be increased in psoriatic lesions [48]. Taken together, our results imply that LL-37 itself induces VEGF-A and COX-2 from keratinocytes and other cells in psoriasis lesions and may be involved in the pathogenesis of psoriasis.

In contrast, the addition of a combination of LL-37 and poly(I:C), a synthetic dsRNA that mimics viral RNA, to keratinocytes induced IFN-β, CXCL-10, and TNF-α, all of which are related to immune defense responses. Type I IFNs such as IFN-β are cytokines that act to suppress viral infection. Viral nucleic acids in infected host cells are detected by cytoplasmic nucleic acid receptors such as the TLR family members RIG-I and MDA5, followed by induction of type I IFNs [49]. Poly(I:C) alone had a minimal effect on keratinocytes at the concentration used in our experiments. In contrast, combined with LL-37, poly(I:C) was incorporated into keratinocytes and efficiently promoted type I IFN responses. LL-37 augments the antiviral activity induced by dsRNA in keratinocytes [50]. These results indicate that LL-37 has both pro-inflammatory and antiviral functions by transporting extracellular nucleic acids into the cells. The exacerbation of skin psoriasis and systemic manifestation of psoriasis are often observed in patients recovering from infectious diseases such as the common cold [51]. Our results suggest that LL-37 is involved in this mechanism.

Scavenger receptors were initially identified by their ability to recognize modified lipoproteins, but are now known to have a wide range of functions, such as the transportation of lipids and intracytoplasmic cargo and the removal of pathogens [52]. In a previous report, we showed that scavenger receptors are involved in the uptake of LL-37 and dsRNA complexes [29]. However, it is not clear if LL-37 alone or the combination of LL-37 and dsRNA facilitate their function using the same receptor and signaling pathways or if they use other specific pathways by which they exert immunomodulatory effects. Therefore, we explored the receptors for LL-37 itself. Of the previously reported receptors for LL-37, FPR2 was at least partially involved in the signal transduction of cathelicidin, since the inhibition of FPR2 partially inhibited the induction of *VEGFA* and *PTGS2* in our experiments. In addition, we newly showed that LL-37 alone as well as the combination of LL-37 and dsRNA binds to multiple types of scavenger receptors, and that the silencing of scavenger receptors by siRNA reduces the number of cytokines induced by LL-37 alone. However, cytokine induction by combined stimulation with LL-37 and poly(I:C) was partially suppressed by the knockdown of a single scavenger receptor, in contrast to previous reports [29,53], probably because multiple scavenger receptors share functional properties [52] and act complementarily on LL-37 uptake. The inhibition of only one scavenger receptor might result in the binding of poly(I:C) and LL-37 to other scavenger receptors. In our experiments, FPR2 inhibitors only partially attenuated LL-37 signaling, whereas the inhibition of multiple scavenger receptors (using fucoidan) almost completely inhibited LL-37 signaling, implying that scavenger receptors are also involved in FPR2-mediated signaling. Furthermore, there may be other receptors for LL-37 that bind to LL-37 and signal through uptake from other scavenger receptors in addition to FPR2.

DsRNA produced a stronger response than dsDNA in combination with LL-37 in NHEKs. However, Nakagawa et al. showed that inflammation induced by CpG ODN was weaker in BMDCs of cathelicidin antimicrobial peptide (*Camp*) knockout mice than in control mice [54], suggesting that the enhancement of the immune response to nucleic acids by LL37 depends on the cell type and animal species. This difference is thought to be due to differences in the expression of nucleic acid receptors by cell type, but since TLR3, 7-9, which recognize nucleic acids, are all expressed on the endoplasmic reticulum in cells, it is possible that differential expression of scavenger receptors may be partly responsible for this difference. 

As in previous reports [29], cytokine induction by the combination of poly(I:C) and LL-37 was reduced by inhibiting clathrin-dependent endocytosis. We newly showed that cytokine induction by LL-37 alone was also reduced by the inhibition of clathrin-dependent endocytosis, probably following the uptake of LL-37 via scavenger receptors. When the transfection reagent Lipofectamine 3000 was used instead of LL-37 in combination with poly(I:C), it induced *IL6* as well as LL-37, but the combination with LL-37 was more effective. The stronger effect was probably due to the fact that LL-37 acted via endocytosis as well as membrane permeabilization. 

Furthermore, LL-37 and poly(I:C) were recognized by endosomal TLRs and the cytoplasmic RNA sensors RIG-I/MDA5/MAVS as previously reported. Knockdown of both TLR3 and MAVS using siRNA almost completely inhibited cytokine induction by the combination of LL-37 and poly(I:C), indicating that both receptors were involved. Kulkarni et al. introduced the term “innate immune vetting” to describe the ability of peptides such as LL-37 to enable certain nucleic acids to act as inflammatory stimuli through scavenger receptor binding prior to cell internalization [53]. In addition, the expressions of TLR3 and RIG-I were increased, indicating that TLR3 and RIG-I signals actually function to induce inflammation. Our results show clearly that the efficient enhancement of the immune response occurs via these pathways.

In this study, the effects of LL-37 on keratinocytes were examined by comparing the effects of LL-37 alone and in combination with poly(I:C). The limitations of this study are that the experiments were limited to keratinocytes and in vitro responses. The mouse cathelicidin mCRAMP does not enhance the poly(I:C) response, unlike LL-37, although mCRAMP has antimicrobial activity [55]. It is therefore difficult to perform experiments regarding the immunological functions of LL-37 using mice. Furthermore, it has been speculated that multiple types of scavenger receptors complement each other in function [52]. While scavenger receptor function is important to the organism, this point makes verification challenging, because scavenger receptor function is difficult to completely suppress. In addition, it is still unclear how LL-37 triggers an immune response after its internalization into the cytoplasm via the scavenger receptors.

Inhibiting the inflammation induced by cathelicidin is difficult due to its diverse functions and its abundant presence in vivo. Our results suggest that LL-37 is not only an amplifier of dsRNA-mediated inflammation, but also a primer for a variety of immune responses. Furthermore, not only immune responses to the combination of dsRNA and LL-37, but also the signaling of LL-37 itself was suppressed by scavenger receptors and endocytosis inhibition. In other words, we showed that the broad function of cathelicidin is commonly dependent on scavenger receptors. Therefore, inhibitors of scavenger receptors or non-functional mock cathelicidin peptides may have utility as new anti-inflammatory and immunosuppressive agents.

## 4. Materials and Methods

### 4.1. Reagents

Synthetic LL-37 peptides were synthesized by and purchased from Genemed Synthesis Inc. (San Antonio, TX, USA). High molecular weight poly(I:C) (tlrl-pic), class C CpG ODN (ODN 2395, tlrl-2395), and poly(dA:dT) (tlrl-patn) were purchased from Invivogen (San Diego, CA, USA). Bafilomycin A1 from *Streptomyces griseus* (B1793) and fucoidan from fucus vesiculosus (F5631) were purchased from Sigma-Aldrich (St. Louis, MO, USA). Pitstop 2^TM^ (ab120687) was purchased from Abcam (Cambridge, UK). Boc-MLF (3730) and WRW4 (2262) were purchased from R&D Systems Inc. (Minneapolis, MN, USA). Lipofectamine 3000 reagent (L3000015) and Lipofectamine RNAiMAX reagent (13778150) were purchased from Thermo Fisher Scientific (Waltham, MA, USA).

### 4.2. Antibodies

The antibodies used in this study are shown in Appendix A.

### 4.3. Culture of Human Primary Keratinocytes

Primary normal human epidermal keratinocytes (NHEKs; neonatal) were cultured in serum-free EpiLife medium with 60 μM calcium (MEPI500CA), EpiLife defined growth supplements (S0125), and antibiotic-antimycotic (100×) (15240062). NHEKs were passaged with trypsin/EDTA solution (R001100) and defined trypsin inhibitor (R007100) before reaching 80% confluence. Passage 3–5 cells at 60–80% confluence were used for the experiments. All items were purchased from Thermo Fisher Scientific. In experiments not otherwise noted, LL-37 was used at 2.5 μM and poly(I:C) was administered 10 min later at 0.3 μg/mL. For inhibitors, WRW4 (10 μM), Boc-MLF (10 μM), KN-62 (10 μM) and Bafilomycin A1 (50 nM) were added 1 h before stimulation, Pitstop 2 (5 μM) 30 min before stimulation, and fucoidan (10 μg/mL) simultaneously with stimulation. Lipofectamine 3000 reagent was pre-mixed according to the product protocol, 1.5 μL of Lipofectamine 3000 reagent per 1 μg of poly(I:C) and 2 μl of p3000 reagent, and added to the cells.

### 4.4. DNA Microarray

NHEKs were stimulated with LL-37 and/or poly(I:C) for 6 h. Total RNA was purified by using an RNeasy mini kit (Qiagen, Venlo, NLD, The Netherlands). The total RNA concentration was measured using a NanoDrop spectrophotometer (Thermo Fisher Scientific). The RNA quality was determined using an RNA 6000 nano kit and an Agilent 2100 bioanalyzer (Agilent Technologies, Santa Clara, CA, USA), and the RNA integrity number was confirmed to be ≥8. Total RNA was amplified, labeled, and analyzed as previously described [56].

GeneSpring GX 14.9 (Agilent Technologies) was used to analyze the expression data of 41,000 genes. After data transformation to GeneSpring, per-chip normalization to the 75th percentile was performed. Extremely low intensity probes were excluded, and probes detected with the flags ‘compromised’ or ‘not detected’ in multiple groups were excluded, leaving 25,875 probes for analysis.

### 4.5. siRNA-Mediated Silencing of Gene Expression

NHEKs were transfected with siRNAs directed against SCARB1 (Dharmacon, Lafayette, CO, L-010592-00), SCARA3 (Santa Cruz Biotechnologies, Dallas, TX, USA, sc-77804), CD68 (Santa Cruz Biotechnologies, sc-35019), Ox-LDL R-1 (Santa Cruz Biotechnologies, sc-40185), RAGE (Santa Cruz Biotechnologies, sc-36374), TLR3 (Dharmacon, L-007745-00), or MAVS (Dharmacon, L-024237-00) at a final concentration of 10 nM, as previously described [29]. The same concentration of ON-TARGET plus non-targeting control pool (Dharmacon, D-001810-10) was used as a control. Briefly, siRNA and Lipofectamine RNAiMAX reagent were prepared in complete antibiotic-free Epilife medium and incubated for 5 min at room temperature. The siRNA and lipofectamine solutions were mixed gently and incubated at room temperature for 20 min to form siRNA-lipofectamine complexes. The siRNA was added to the cell culture medium, which contained a final concentration of 0.25% lipofectamine. After 24 h of transfection, the medium was changed to complete Epilife medium with antibiotics. After 24 h of incubation, the cells were treated with LL-37 and poly(I:C) for 6 h, then collected.

### 4.6. Quantitative RT-PCR Analysis

The extraction of total RNA and quantification of RNA were conducted in the same manner as described in the DNA microarray section. cDNA was generated using up to 1 μg of total RNA by reverse-transcription using a PrimeScript™ RT-PCR kit from Takara Bio (Shiga, Japan). Quantitative real-time PCR was performed using an AriaMx Real-Time PCR System (Agilent Technologies) with Brilliant III Ultra-Fast QPCR Master Mix (Agilent Technologies) and Taqman^®^ Gene Expression Assays (Thermo Fisher Scientific). Primers and probes for GAPDH were prepared based on known sequences. PCR primers and probes are shown in Appendix A.

### 4.7. Enzyme-Linked Immunosorbent Assay (ELISA)

The cell culture supernatant was frozen at −80 °C until use for the analysis.

The levels of VEGF-A and CXCL-10 were measured using a Human VEGFA DuoSet ELISA kit (R&D Systems Inc., DY293B) and Human CXCL10/IP-10 DuoSet ELISA kit (R&D Systems Inc., DY266), respectively.

### 4.8. Western Blotting

The samples derived from NHEKs were lysed in RIPA buffer (Cell Signaling Technology, 9806S) with phenylmethylsulfonyl fluoride (PMSF), or in a denaturing lysis buffer containing 20 mM HEPES pH 7.4, 250 mM NaCl, 2 mM EDTA, and 1% SDS supplemented with completed proteinase inhibitor cocktail as well as 50 mM sodium fluoride, 5mM N-ethylmaleimide, and 100 μM hemin chloride to maximally preserve protein post-translational modifications as described previously [57]. After sonication and centrifugation, protein concentrations were determined using a Pierce™ BCA protein assay kit (Thermo Fisher Scientific, 23227). Ten to twenty micrograms of protein were electrophoresed with a Mini-PROTEAN Precast Gel (Bio-Rad Laboratories Inc., Hercules, CA, USA) and transferred to an Immobilon^®^-P PVDF Membrane (Millipore, Billerica, MA, USA) or Trans-Blot Turbo Mini-size LF PVDF membrane (Bio-Rad Laboratories Inc.) using a Trans-Blot Turbo system (Bio-Rad Laboratories Inc.). Chemiluminescence immunoblotting was performed using primary antibodies and HRP-conjugated secondary antibodies, followed by a luminescence assay with LumiGLO^®^ and a peroxide reagent (Cell Signaling Technology, 7003) and imaging on an ImageQuant LAS 4000 mini (GE Healthcare, Pittsburgh, PA, USA). Fluorescence immunoblotting was performed with primary antibodies and an IRDye 800 CW secondary antibody (LI-COR Biosciences, Lincoln, NE, USA), followed by imaging using an Odyssey imaging system (LI-COR Biosciences).

### 4.9. Proximity Ligation Assay (PLA)

NHEKs were seeded in 8-well chamber slides (MATSUNAMI, Osaka, Japan) and incubated without stimulation or with LL-37 alone or in combination with poly(I:C) for 1 h at 4 °C to prevent internalization. Cells were washed with cold PBS and fixed in cold 4% paraformaldehyde (PFA). After blocking according to the manufacturer’s instructions (Sigma-Aldrich), cells were incubated with two primary antibodies. Secondary antibodies conjugated with oligonucleotides were added, then ligation and amplification were performed to generate a fluorescent signal in the region where the antigen recognized by the two primary antibodies was present below 40 nm. Fluorescent PLA signals were detected and photographed with a laser scanning confocal microscope (LSM700; ZEISS, Tokyo, Japan). PLA signal counts were measured with ImageJ software (Fiji; version 2.1.0; National Institutes of Health, Bethesda, MD, USA).

### 4.10. Statistical Analysis

All statistical analyses were performed with GraphPad Prism 9 software (Dotmatics, Bishop’s Stortford, UK, version 9.0.0). To compare means between more than two groups, two-way analysis of variance (ANOVA) with Bonferroni’s post hoc test was performed. A value of *p* < 0.05 was considered significant, with *p* > 0.05, ** *p* ≤ 0.01, *** *p* ≤ 0.001, **** *p* ≤ 0.0001. No statistical methods were used to predetermine sample size. The experiments were not randomized. The investigators were not blinded.

## Figures and Tables

**Figure 1 ijms-24-00875-f001:**
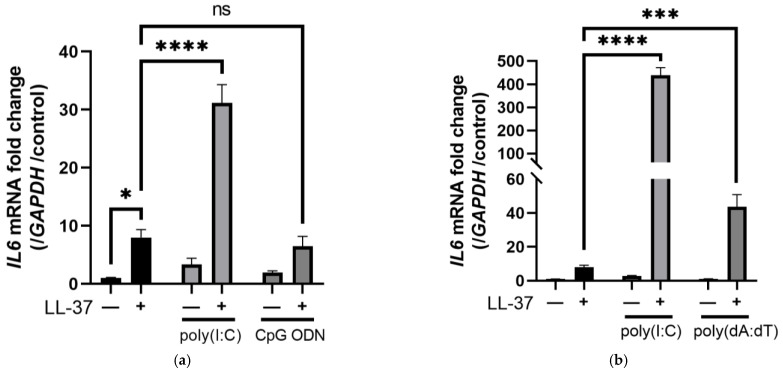
DNA microarray revealed that LL-37 alone induced biological and metabolic processes, while the combination of LL-37 and poly(I:C) induced immune responses in keratinocytes. (**a**) Poly(I:C) (0.3 μg/mL), CpG ODN (14.1 μg/mL) or (**b**) poly(dA:dT) (1 μg/mL) with LL-37 (2.5 μM) each alone or in combination were added to NHEK and cultured for 6 h. The induction of *IL6* mRNA was measured by RT-PCR. NHEKs were stimulated with LL-37 at different concentrations (0.313–5 μM) alone or in combination with poly(I:C). RT-PCR was performed to quantify the mRNA expressions of (**c**) *IL6*, (**d**) *IL36G*, and (**e**) *IFNB1*. NHEKs were stimulated with LL-37 and poly(I:C) alone or in combination and analyzed by DNA microarray. (**f**) A Venn diagram shows the number of genes upregulated by each stimulation. Blue: LL-37, green: poly(I:C), red: LL-37+poly(I:C). Gene ontology analyses of genes induced when stimulated with (**g**) LL-37 alone or (**h**) in combination with poly(I:C) are shown. (**i**) Summary of the genes shown in Figure 1f that were increased by LL-37 stimulation but not by other stimuli (blue) and genes that were increased only by co-stimulation with LL-37 and poly(I:C) (red). Data are means ± SEM of three biological replicates. NS, *p* > 0.05, * *p* ≤ 0.05, *** *p* ≤ 0.001, **** *p* ≤ 0.0001 by two-way ANOVA with Bonferroni’s post hoc test. See also Appendix A.

**Figure 2 ijms-24-00875-f002:**
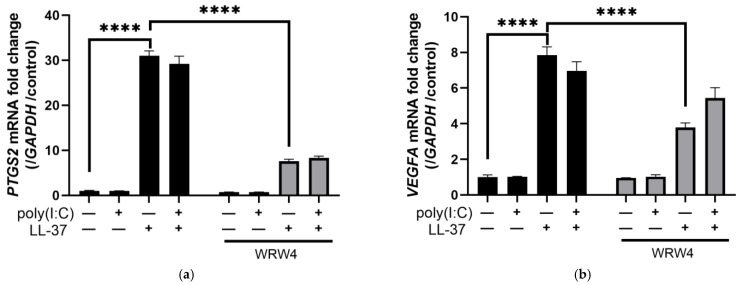
The FPR2 inhibitor WRW4 inhibited the induction of *PTGS2*, *VEGFA*, and phosphorylated p38 induced by LL-37 stimulation in keratinocytes. NHEKs were stimulated with LL-37 and poly(I:C) alone or in combination in the presence or absence of WRW4 (a FPR2 inhibitor), and RT-PCR was performed. Fold changes in (**a**) *PTGS2*, (**b**) *VEGFA*, (**c**) *CXCL10*, and (**d**) *IFNB1* mRNA are shown. NHEKs were stimulated under the same conditions for 6 h, and culture supernatants were analyzed by ELISA. Protein concentrations of (**e**) VEGF-A and (**f**) CXCL-10 are indicated. RT-PCR was conducted by stimulating NHEKs with poly(I:C) and LL-37 alone or in combination in the presence or absence of Boc-MLF (a FPR1 inhibitor) or KN-62 (a P2X7R antagonist). Fold changes in (**g**) *PTGS2*, (**h**) *VEGFA*, (**i**) *CXCL10*, and (**j**) *IFNB1* mRNA are shown. (**k**) NHEKs were pretreated with WRW4 for 1 h or Pitstop 2 for 30 min and stimulated with LL-37 and then poly(I:C) 10 min later. Cells were collected after 30 min, and protein expressions of p-p38 and p38 were assessed by immunoblots. β-actin was used as a loading control. Data are means ± SEM of three biological replicates. NS, *p* > 0.05, * *p* ≤ 0.05, ** *p* ≤ 0.01, *** *p* ≤ 0.001, **** *p* ≤ 0.0001 by two-way ANOVA with Bonferroni’s post hoc test. See also Appendix A.

**Figure 3 ijms-24-00875-f003:**
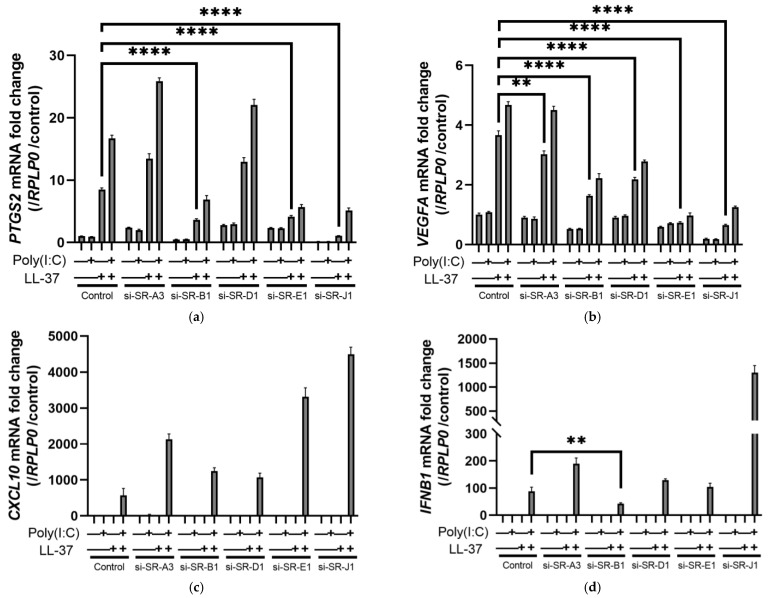
Stimulation by LL-37 alone or in combination with poly(I:C) in NHEKs is mediated by scavenger receptors. SCARA3 (SR-A3), SCARB1 (SR-B1), CD68 (SR-D1), OLR1 (SR-E1), AGER (SR-J1), and non-targeting pool siRNAs were transfected into NHEKs, and the cells were stimulated with LL-37, poly(I:C) or LL-37+poly(I:C) 2 days later. After 6 h of incubation, mRNA induction was evaluated by RT-PCR. The inductions of (**a**) *PTGS2*, (**b**) *VEGFA*, (**c**) *CXCL10,* (**d**) *IFNB1,* and (**e**) *TNFA* are shown, respectively. (**f**) PLA was performed for various scavenger receptors and LL-37. LL-37 was added to NHEKs, 10 min later poly(I:C) was added, and the cells were incubated at 4 °C for 1 h. Using antibodies against various scavenger receptors and LL-37, the proximity of each scavenger receptor and LL-37 was detected by fluorescence-based PLA as green fluorescent spots. Nuclei (blue) were counterstained with DAPI. Data are means ± SEM of three biological replicates. ** *p* ≤ 0.01, **** *p* ≤ 0.0001 by two-way ANOVA with Bonferroni’s post hoc test. See also Appendix A.

**Figure 4 ijms-24-00875-f004:**
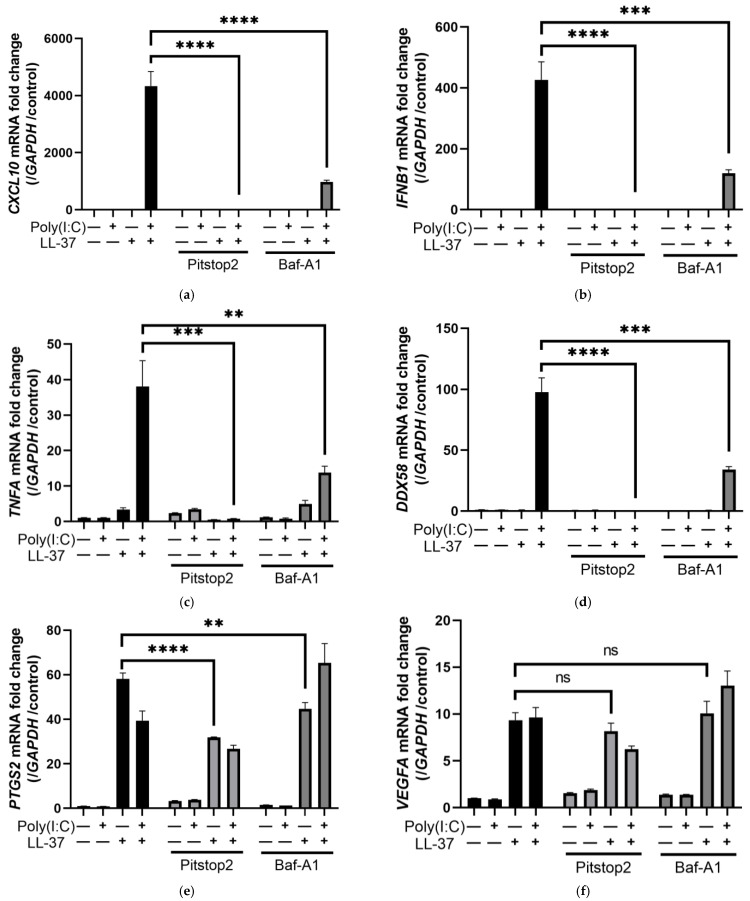
Co-stimulation of LL-37 and poly(I:C) in NHEKs induced various cytokines via clathrin-dependent endocytosis. NHEKs were pretreated with Baf-A1 (an inhibitor of endosomal receptors including TLR3) for 1 h or Pitstop 2 (a clathrin-dependent endocytosis inhibitor) for 30 min, then stimulated with LL-37 and poly(I:C) alone or in combination. Relative mRNA expressions of (**a**) *CXCL10*, (**b**) *IFNB1*, (**c**) *TNFA*, (**d**) *DDX58*, (**e**) *PTGS2*, and (**f**) *VEGFA* were quantified by RT-PCR. NHEKs were stimulated under the same conditions, and the protein levels of (**g**) CXCL-10 and (**h**) TNF-α in culture supernatants were analyzed by ELISA. NHEKs were transfected with siRNAs of TLR3, MAVS, or a combination of TLR3 and MAVS, and stimulated with LL-37 and poly(I:C) 2 days later. Relative mRNA expressions of (**i**) *CXCL10*, (**j**) *IFNB1,* and (**k**) *TNFA* were quantified by RT-PCR. (**l**) NHEKs were pretreated with Pitstop 2 or WRW4 (an FPR2 inhibitor) and stimulated with poly(I:C) and LL-37, and 2 h later the cells were collected. Protein levels of p-IRF3, IRF3, p-TBK1, and TBK1 were evaluated by immunoblotting. β-actin was used as a loading control. Data are means ± SEM of three biological replicates. NS, *p* > 0.05, * *p* ≤ 0.05, ** *p* ≤ 0.01, *** *p* ≤ 0.001, **** *p* ≤ 0.0001 by two-way ANOVA with Bonferroni’s post hoc test. See also Appendix A.

## Data Availability

Not applicable.

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
