# Peer review of "The Antimicrobial Peptide Cathelicidin Exerts Immunomodulatory Effects via Scavenger Receptors"

_ijms, 2023, doi:10.3390/ijms24010875_

Round 1

Reviewer 1 Report

This paper described the effect of LL-37 alone, and LL-37 with poly I:C on gene expression and signaling pathways in NHEKs. The current is well designed, but there are some comments.

 This paper focused on the LL-37, but seems to be missing the proper controls. In particular, poly I:C (high concentration) alone, and poly I:C mixed with a transfection agent. In addition, dsDNA and ssRNA as a DAMPS are also needed.

 In this paper, two treatment conditions, multiple gene expression, and several steps have been described in complexity. It is hard to understand. Therefore, reviewers are suggested to present a graphical summary.

Reviewer 2 Report

This paper analyses the individual and combined effects of the antimicrobial peptide LL37 and poly(I:C) a TLR3 agonist on innate immune signaling in keratinocytes.

While many of the results are of interest, the manuscript relies too heavily on the use of pharmacological inhibitors. Importantly, the role of TLR3 signaling, which may be central, is not firmly established in the study in its present form. Part of this stems from a failure to knockdown TLR3 signaling using siRNA, which was not efficacious under the conditions used. The authors state that the knockdown was not efficacious because TLR3 transcripts were induced by LL37 and poly(I:C). This does not make much sense because the knockdown should have occurred prior to addition of the stimuli. This experiment should be redone. It is important because there are multiple nucleic acid-sensing TLRs (e.g. TLRs 3, 7-9) and it is important to determine which one(s) are active. It may be necessary to leave cells in culture longer than 24h to achieve efficacious knockdown, depending on the rate of turnover of TLR3 protein. This should be followed by western blotting to optimize conditions.

In addition, TLR3 phosphorylation on tyrosine (e.g. Y759) is induced upon ligand stimulation. As commercially available antibodies are available against TLR3 pY759, the authors should examine individual and combined effect of poly(I(I:C) and LL37 on this process.

Round 2

Reviewer 1 Report

No more comments.

Reviewer 2 Report

The authors have now responded adequately to my main concerns.